# Early Postnatal Outcome and Care after in Utero Exposure to Lithium: A Single Center Analysis of a Belgian Tertiary University Hospital

**DOI:** 10.3390/ijerph191610111

**Published:** 2022-08-16

**Authors:** Marlien Torfs, Titia Hompes, Michael Ceulemans, Kristel Van Calsteren, Christine Vanhole, Anne Smits

**Affiliations:** 1Department of Pediatrics, University Hospitals Leuven, 3000 Leuven, Belgium; 2Mind-Body Research Unit, Department of Neurosciences, KU Leuven, 3000 Leuven, Belgium; 3Adult Psychiatry UPC, KU Leuven, 3000 Leuven, Belgium; 4L-C&Y, Child and Youth Institute, KU Leuven, 3000 Leuven, Belgium; 5Clinical Pharmacology and Pharmacotherapy, Department of Pharmaceutical and Pharmacological Sciences, KU Leuven, 3000 Leuven, Belgium; 6Teratology Information Service, Netherlands Pharmacovigilance Centre Lareb, 5237 MH Hertogenbosch, The Netherlands; 7Department of Development and Regeneration, KU Leuven, 3000 Leuven, Belgium; 8Department of Obstetrics and Gynecology, University Hospitals Leuven, 3000 Leuven, Belgium; 9Neonatal Intensive Care Unit, University Hospitals Leuven, 3000 Leuven, Belgium

**Keywords:** lithium, in utero exposure, neonate, postnatal care

## Abstract

Knowledge of the impact of in utero exposure to lithium during the postnatal period is limited. Besides a possible teratogenic effect during the first trimester, exposure during the second and third trimesters might lead to neonatal effects. Uniform guidelines for postnatal management of these neonates are lacking. The aim was to retrospectively describe all neonates admitted to the University Hospitals Leuven after in utero exposure to lithium (January 2010 to April 2020), and to propose a postnatal care protocol. Descriptive statistics were performed. For continuous parameters with serial measurements, median population values were calculated. In total, 10 mother-neonate pairs were included. The median gestational age was 37 (interquartile range, IQR, 36–39) weeks. Neonatal plasma lithium concentration at birth was 0.65 (IQR 0.56–0.83) mmol/L with a median neonate/mother ratio of 1.02 (IQR 0.87–1.08). Three neonates needed respiratory support, 7/10 started full enteral (formula) feeding on day 1. The median length of neonatal stay was 8.5 (IQR 8–12) days. One neonate developed nephrogenic diabetes insipidus. This study reported in detail the postnatal characteristics and short-term neonatal outcomes. A postnatal care protocol was proposed, to enhance the quality of care for future neonates, and to guide parental counselling. Future prospective protocol evaluation is needed.

## 1. Introduction

Bipolar disorder is a severe psychiatric condition occurring with a lifetime prevalence of 1–2.4% [1,2]. The onset of the disease is around the age of 20 years, which means that women are commonly affected during their reproductive period [1,2,3]. During pregnancy and the postpartum period, the risk of a symptomatic exacerbation is elevated [4]. Pharmacotherapy for bipolar disorder (anti-epileptic drugs, lithium) during this period needs to be based on a risk/benefit analysis for both mother and fetus/child, as mentioned in different guidelines [5,6,7]. Anti-epileptics like valproate and carbamazepine are often avoided due to teratogenicity. Lithium is the first-choice treatment for bipolar disorder. It reduces manic and depressive relapse, and suicide risk [8,9]. Also during pregnancy and the postpartum period, lithium is proven effective, and is often used (albeit cautiously) as maintenance therapy, or to prevent postpartum psychosis [9,10,11]. Mechanisms regulating cell membrane properties, cell membrane transport, neurotransmitter regulation, ion distribution, and intracellular signalling may contribute to the mood-stabilizing action of the drug [12,13]. An important characteristic of lithium is its equilibration across the placenta, reported by Newport et al. with a mean infant-mother lithium ratio at delivery of 1.05 (SD 0.13) [14].

Of all psychotropic drugs, lithium has the most clearly documented teratogenic effect [15]. Although the International Register of Lithium Babies [16] suggested more than 30 years ago a marked increase in cardiac malformations including Ebstein’s anomaly, Patorno et al. later attenuated the magnitude of this effect [17]. More recently, the likely association of lithium use during pregnancy with cardiac malformation was also reported in a meta-analysis, and a large population-based cohort study, but this risk indeed remains low [10,18]. In contrast, the meta-analysis of Munk-Olsen et al. documented a significantly increased risk for major malformations, but not for cardiac malformations [15].

Knowledge on the impact of in utero exposure to lithium during the postnatal period is limited and mainly based on case reports or small cohorts [19,20,21,22,23]. In addition to the possible teratogenic effect associated with lithium in the first trimester, exposure during the second and third trimesters might lead to other complications [6]. Besides the increased risk of respiratory problems during the adaptation phase after birth, electrolyte disturbances, nephrogenic diabetes insipidus (NDI), hypoglycemia and thyroid dysfunctions have also been reported in the neonate [19,20,21]. Based on a retrospective analysis, Molenaar et al. recently reported on a neonatal cohort with in utero exposure to lithium, of whom most mothers had lithium levels within the therapeutic window (0.5–1.2 mmol/L) [9,23]. The authors observed no association between neonatal lithium blood levels at delivery and neonatal outcomes [23].

Today, uniform guidelines for medical postnatal management of the neonates exposed to maternal lithium (e.g., timing and frequency of blood sampling for electrolytes, lithium therapeutic drug monitoring (TDM), …) are lacking. To improve the postnatal care of this specific neonatal population, the aim was to retrospectively review and describe the cases admitted for this indication to the neonatal department of the University Hospitals Leuven over the past 10 years and subsequently propose a medical postnatal care protocol. 

## 2. Materials and Methods

### 2.1. Study Population and Data Collection

In this retrospective study, all neonates with in utero exposure to lithium admitted to the neonatal intensive care unit (NICU) or medium care unit of the University Hospitals Leuven, Belgium, from January 2010 until April 2020, were included. The study was approved by the Ethics Committee of the University Hospitals Leuven (study ID S64062, approval date 5 June 2020). Eligible mother-neonate pairs were identified by two routes. First, available maternal and neonatal lithium TDM results determined during the study period were obtained from the hospital laboratory records. Second, medical files of women receiving a lithium prescription at the obstetrics or maternity ward in the study period were reviewed. Neonatal and corresponding maternal cases from whom sufficient data could be retrieved were included as mother-neonate pairs. Duplicates from the 2 searches, were removed. Clinical and laboratory data were extracted from medical records. Collected neonatal data at birth were sex (male, female), gestational age (GA, weeks), birthweight (BW, grams), APGAR scores, umbilical pH at delivery and presence of congenital malformations. Collected hospitalization data were respiratory and feeding evolution, Finnegan scores, need for neonatal abstinence syndrome (NAS) treatment, daily urine output (mL/kg/h), length of hospital stay (days), and specific biochemical data including plasma sodium (mmol/L), plasma potassium (mmol/L), total and direct bilirubinemia (mg/dL), creatininemia (mg/dL), a liver panel consisting of aspartate transaminase (AST, U/L) and alanine transaminase (ALT, U/L), glycemia (mg/dL), lithium TDM (mmol/L) and thyroid function consisting of thyroid stimulating hormone (TSH, mIU/L) and free thyroxine (FT4, pmol/L). The biochemical data were real-world data, collected during routine clinical care. No formal protocol on biochemical assessment after in utero exposure to lithium is present in our unit, except for the fact that a first lithium TDM is usually collected on the day of birth (i.e., cord blood or venous sampling). The day of birth was defined as postnatal age (PNA) day 1. Collected maternal data were medical diagnosis, relevant co-morbidities, lithium dose at delivery, lithium TDM values, concomitant medication and pregnancy complications.

### 2.2. Lithium Assay

Lithium concentrations in serum were determined with a colorimetric endpoint reaction. From 2010–2012, Dimension RxL (Siemens, Siemens Healthcare, München, Germany) was used, from 2012 Cobas 8000 (Roche, Roche Diagnostics, Mannheim, Germany) is used with a current lower limit of quantification of 0.05 mmol/L.

### 2.3. Statistical Analysis

Descriptive statistics were performed and data were presented as median (interquartile range, IQR) or incidence. For continuous parameters with serial measurements (e.g., daily diuresis, plasma sodium, …) median values of the study population were calculated and presented graphically over time using boxplots. Analysis was performed using Excel (Microsoft Corporation, Redmond, Washington, DC, USA, Microsoft 365 version 2204) and Medcalc (MedCalc^®^ Statistical Software version 20.110, MedCalc Software Ltd., Ostend, Belgium). For figures, Medcalc and SPSS (IBM SPSS Statistics, version 25, IBM Corp., Armonk, NY, USA) were used. 

## 3. Results

### 3.1. Study Population

Based on the lithium TDM search, 14 mothers and 7 neonates were retrieved. The search based on lithium prescriptions led to 15 mothers and their 15 respective neonates. After matching the mother-neonate pairs, and after removing duplicates, cases missing sufficient maternal or neonatal data were excluded. With 19 mother-neonate pairs left, 9 cases in which maternal lithium therapy was interrupted during pregnancy were excluded. Finally, the study cohort consisted of 10 mother-neonate pairs (Figure 1).

Maternal characteristics at delivery are presented in Table 1. The maternal diagnosis for each case was bipolar disorder (*n* = 10). Median maternal TDM at delivery was 0.62 mmol/L (IQR 0.54–0.72 mmol/L). Besides hypertension (*n* = 1) and intrahepatic cholestasis of pregnancy (*n* = 1) as pregnancy complications, 4 cases suffering from gestational diabetes mellitus (2 of them insulin-dependent) were defined. The use of concomitant medication was present in 9/10 cases. The most frequently used medicine was levothyroxine (*n* = 7). One mother was co-treated with an anti-epileptic drug (lamotrigine), and 6 mothers with antipsychotic medication (1 conventional antipsychotic: haloperidol (*n* = 1), 3 atypical antipsychotics: quetiapine (*n* = 3), olanzapine (*n* = 2) and aripiprazole (*n* = 1)), and 1 mother with the benzodiazepine lorazepam during their pregnancy (Table 1). 

Neonatal characteristics at birth and during hospital stay are provided in Table 2 and Table 3, respectively. The neonatal cases consisted of 6 female and 4 male neonates. The majority were born full term (*n* = 7). Overall median GA was 37 (IQR 36–39) weeks. Of the 3 preterm cases (GA < 37 weeks), 1 was very preterm (GA < 32 weeks), and 2 were late preterm (GA > 34 weeks). With a median Apgar score of 9 at 5 min, most babies did not need further intervention immediately after birth. Cord blood analysis showed a median arterial umbilical pH of 7.23 (IQR 7.17–7.27). Median BW was 3000 g (IQR 2620–3440), with one neonate large for GA (BW > 90th percentile) and small for GA (BW < 10th percentile). One neonate presented with a congenital malformation, more specifically a congenital diaphragmatic hernia (CDH) (Table 2). This case was also diagnosed with NDI, with a maximum diuresis of 16 mL/kg/h (Figure 2). Further investigation consisted of a renal ultrasound (two normal kidneys), and a desmopressin test to exclude central diabetes insipidus. Besides one observed diuresis of 6.07 mL/kg/h limited to PNA day 2 in one neonate, diuresis of all other neonates remained below a maximum observation of 4.65 mL/kg/h.

Concerning the neonatal characteristics during hospital stay (Table 3), 3 neonates needed respiratory support. Two needed continuous positive airway pressure (CPAP) for 1 day, and 1 neonate needed mechanical ventilation during its hospital stay. All neonates (*n* = 10) were formula fed, 7 of them started with full enteral feeding on the day of birth. After 7 days, only 1 neonate was not yet on full oral feeding. This was the neonate with the CDH. Surgical repair of the CDH was performed at PNA day 5. At PNA day 37, this neonate was transferred to another hospital, closer to the parents’ home. 

In Figure 3, the incidence of cases with in utero exposure to lithium was plotted for each year of the study period. Most cases (*n* = 9) presented during the last 5 years. During the first 6 years, only 1 case was included. 

### 3.2. Biochemical Data

Most neonates had an elaborated blood panel on day 1, of which the results are summarized in Table 4. On day 1, plasma sodium, plasma potassium, creatininemia and FT4 for almost every neonate (*n* = 9) were available. While glycemia and TSH were also routinely determined (*n* = 10), liver panels on day 1 were rather scarce (*n* = 3). One neonate was hypoglycemic on day 1, with a lowest glycemia of 20 mg/dL. Concerning available neonatal biochemical data after day 1, the timing of routine blood sampling varied between the included cases (no structured guidance on PNA and on sampling parameters). Available data on total and direct bilirubinemia, creatininemia, plasma potassium, plasma sodium, ALT, AST, FT4, and TSH of the study population during the neonatal period are provided as Appendix A respectively.

### 3.3. Neurological Assessment and Lithium TDM

The clinical neurological short-term outcome of the neonates was assessed. For 6 cases, Finnegan scores were available from the medical files. Values ranged from 0–7, with the highest values measured during the first 3 days. The highest daily scores during the first week of life showed a declining trend in 5 out of 6 cases. Clinical neurological examinations and neonatal/maternal lithium TDM ratio at birth were assessed using nursing and medical files (Table 5). The examination was normal for 5 neonates. Neonatal lithium TDM on day 1 was available for 8/10 cases, with a median value of 0.65 (0.56–0.83) mmol/L (Figure 4). Median lithium TDM showed a declining trend with PNA (Figure 4). Individual TDM data over time are presented in Appendix A.

## 4. Discussion

### 4.1. Main Findings

The primary aim of this retrospective study was to assess the early postnatal characteristics and short-term outcomes of neonates with in utero exposure to lithium. In total, 10 mother-neonate pairs were included. A few results caught our attention. First, 3 cases in this study population displayed hypotonia (Table 5), 3 had respiratory symptoms (Table 3) and 1 had hypoglycemia (Table 4). Four neonates displayed a (mild) abnormal clinical neurological evaluation in the early postnatal phase (Table 5). According to the literature, neonates with in utero exposure to lithium during the second and third trimesters could possibly present with lithium (adverse) effects. Neonatal lithium-related complications documented in case reports are, for example, thyroid dysfunction, cardiac arrhythmia, hepatic abnormalities, and hypoglycemia [8,9,14,18]. Adverse lithium effects can also present as ‘floppy infant syndrome’ [24]. Lethargy, poor sucking, tachypnea, tachycardia, respiratory distress syndrome, cyanosis and hypotonia are commonly described symptoms in the literature of neonates with in utero exposure to lithium [25]. Although causality could not be assessed due to the small sample size, these findings illustrate that indeed mild to moderate neonatal symptoms after in utero exposure to lithium have to be anticipated.

Second, in this cohort, the Finnegan score was used to evaluate the presence of NAS [26,27,28,29]. However, the score was not used consistently during the study period, leading to missing data. This stresses the need to train nursing and medical staff to systematically apply this score to neonates at risk for NAS. It has been documented that the most effective interventions impacting the length of hospital stay for infants with NAS are the development of a staff NAS education program and the implementation of a treatment protocol [30]. Although initially designed to evaluate NAS severity due to opioid withdrawal [28], the Finnegan score is currently applied to a broader field of in utero drug exposure, due to the absence of compound-specific scoring tools [29]. Systematic scoring allows us to assess NAS severity and evolution, and to decide on initiating pharmacotherapy [29]. 

Third, lithium levels at birth reflect neonatal exposure. The median maternal lithium TDM at delivery in the cohort was 0.62 mmol/L (IQR 0.54–0.72 mmol/L), and the neonatal lithium TDM at birth was 0.65 mmol/L (IQR 0.56–0.83 mmol/L). This illustrates the equilibration of lithium across the placental barrier, with a median neonatal/maternal TDM ratio of 1.02 (IQR 0.87–1.08). Clark et al. reported that a lithium cord level exceeding 0.64 mEq/L has already been associated with several neonatal complications such as respiratory distress and neurological symptoms [31]. According to these findings, the delivery of a neonate with in utero exposure to lithium should be well-planned, preferably in a centre with a dedicated multidisciplinary perinatal psychosocial team with prenatal counselling, carefully monitored maternal lithium TDM, and experience in neonates with this condition [7,8,9]. Some guidelines suggest interrupting lithium therapy at the onset of labour or 24 to 48 h before induction of labour or caesarian section, to lower lithium TDM at birth to prevent neonatal adverse effects [6,31,32]. However, a therapy interruption in this period might lead to a higher risk of relapse during the postpartum period [33,34]. Provided that lithium TDM is strictly monitored during the perinatal period, Molenaar et al. do not recommend anymore the interruption of lithium therapy before delivery based on their recent observational study [23]. 

Fourth, our study population contained one neonate diagnosed with NDI. Based on the literature, this might be induced by lithium exposure during pregnancy. The underlying pathophysiology of NDI is complex and consists of different mechanisms [21,35]. Lithium enters the principal cells of the collecting duct through the epithelial sodium channel (ENaC, competition with sodium). There, it inhibits the action of antidiuretic hormone (vasopressin) [35]. Lithium causes a decrease in intracellular cyclic adenylate cyclase in the principal cells of the collecting tubule resulting in an inhibition of the glycogen synthase kinase-3, and subsequent decreased aquaporin 2 (AQP2) expression, leading to reduced reabsorption of water. This results in polyuria and NDI [36,37]. In addition, decreased sensitivity to vasopressin, increased prostaglandin signalling, and other molecular effects of lithium also cause decreased AQP2 expression, polyuria and NDI [38]. NDI in the neonate after in utero exposure to lithium has previously been described. The most important characteristics of this condition are polyuria, with normal to high serum osmolality and low urinary osmolality [20,21,39]. In our case, a maximum diuresis of 16 mL/kg/h was registered on PNA days 1–2, combined with hypernatremia (150 mmol/L), and a low urinary osmolality of 166 mmol/kg. On the same day, a neonatal lithium TDM of 0.1 mmol/L was measured. The presence of natriuresis and the transient character of the disease help to distinguish primary NDI due to mutations in the AQP2 channel and vasopressin-receptor 2 gene from lithium-induced polyuria. The most important reason to treat this condition in time is to avoid dehydration and further lithium intoxication, by replacing fluid deficits [39]. The same neonate was diagnosed with a CDH, contributing to the need for intensive care treatment, including mechanical ventilation and no full enteral feeding (Table 3). Based on the neonatal characteristics of the study cohort, increased awareness of the need for postnatal monitoring and clinical observation of neonates exposed to maternal lithium is needed. Even with low exposure, neonatal effects may appear. Careful evaluation of diuresis during the first days of life should be an integral part of this follow-up. 

Fifth, the neonates in the study cohort all received formula feeding as lithium is contra-indicated during breastfeeding in different guidelines [6,7,9,32,40]. The drug has a relative infant dose (RID) of 0.87–30% and is classified as lactation risk category 4, indicating there is significant evidence lithium is excreted into human milk and has possible hazardous properties [41]. Lithium excretion into human milk combined with neonatal immature renal function (and subsequently increased half-life compared to adults) may lead to a risk of lithium intoxication [42]. RID values and effects on neonates vary strongly among studies [43]. Due to this high variability, most guidelines consider maternal lithium therapy during postpartum not compatible with breastfeeding. Therefore, also in our unit, formula feeding is used as a standard of care for these cases. In addition, women diagnosed with bipolar disorder are vulnerable to relapse during the postpartum period and continuation of lithium therapy is often needed. Maternal support by caregivers and relatives is important during this period, e.g., to minimize sleep deprivation [42].

However, we are aware that the discussion to breastfeed these neonates or not is a topic of interest and debate in recent literature. Imaz et al. summarized that publications not arguing against the use of lithium during breastfeeding are also available [44]. Based on their systematic review published in 2019 on clinical lactation studies of lithium, 20.5% of breastfed infants with maternal lithium use presented transient short-term adverse effects [44]. The available information was based on a small and heterogeneous number of case reports or case series, and the quality of the studies included was all less than optimal [44]. In 2021, it was documented that the median times for lithium serum concentration to reach a limit of quantification of 0.20 mEq/L in full-term neonates of mothers with lithium monotherapy, and receiving formula, mixed (formula/breastfeeding), or exclusive breastfeeding were 6–8, 7–8, and 53–60 days respectively [45]. Very recently, Heinonen et al. (2022) considered lithium during breastfeeding safe in selected cases, and under strict follow-up [46]. Although insights into the pharmacokinetics of lithium during lactation are increasing [47], further research is warranted.

Finally, the incidence of cases was not evenly distributed over time. Figure 3 shows that lithium use was relatively more frequent during the last five years. A first explanation might be found in a change of clinical guidelines on antenatal management of women with bipolar disorder. Since 2014, the National Institute for Health and Care Excellence has proposed to consider the continuation of lithium during pregnancy in specific conditions [6]. Furthermore, as highlighted earlier, recent studies showed that the association between lithium use and congenital malformations is of a smaller magnitude than previously reported [15,17]. The slightly elevated risk of congenital malformations due to lithium use during the first trimester (4.2%), and the risk of neonatal lithium effects due to exposure in the second and third trimester, should be weighed against the high risk of maternal relapse during pregnancy and postpartum (20 to 70% over 12 months) [10]. Pregnancy is no longer an absolute contraindication to lithium use [6,32]. This may contribute to the higher incidence during the second half of the study period. Second, preconception and prenatal counselling consultations by the perinatal psychiatric team in our hospital started in 2014, resulting in an increase in the number of patients. 

### 4.2. Proposal of a Postnatal Care Protocol for Neonates after In Utero Exposure to Lithium

The secondary aim of this study was to propose a medical postnatal care protocol for neonates with in utero exposure to lithium. Based on the characteristics and short-term outcome of the study cohort, along with literature, a proposal was drafted (Figure 5), which clearly is the subject of further research. Due to the risk of postnatal symptoms, careful observation of clinical evolution, vital signs and (modified) Finnegan scoring is suggested during the first days of life. Depending on the respiratory and general status of the neonate, this can be done at a medium care unit (NICU if respiratory or other NICU support is needed). In literature, rooming-in for infants at risk for NAS from in utero opioid exposure is supported [48,49]. Multidisciplinary assessment of the maternal medical and psychosocial condition is needed for women suffering from bipolar disorder to decide (often case-by-case) if rooming-in is a suitable option. The availability of trained healthcare professionals for scoring, monitoring and supportive management is important. 

Concerning the duration of observation and supportive treatment, Smirk et al. investigated 210 neonates who received NAS treatment after in utero exposure to various drugs, predominantly opioids [50]. Within five days, they detected that 95% of the neonates with NAS symptoms required treatment [50]. Official guidelines on the duration of observation specifically applicable for neonates with in utero exposure to lithium are lacking. Molenaar et al. reported a rather high rate of neonatal complications (48.3%) after in utero exposure to lithium, and also referred to an observation period of 5 days [23]. For a drug with a high toxicity profile, such as lithium, this period seems indeed requested. During hospital stay clinical observation and the following supportive measures can be suggested:
Monitoring cardiorespiratory parameters (heart rate, oxygen saturation, respiration rate)Measurement of diuresis to detect polyuriaEvaluation of (risk for) NAS using the (modified) Finnegan score, every 3 to 4 h.Non-pharmacological supportive management (as part of the general NAS approach) [51]:
Environmental control (swaddling, low-stimulus environment, …) Formula feeding on demandParental education


Besides supportive treatment, biochemical parameters need to be evaluated (Figure 5). Most relevant are lithium TDM, plasma sodium and potassium, glycemia, TSH, FT4, AST, ALT, creatininemia and bilirubinemia (total and direct). Lithium TDM sampling on day 1 can be collected from umbilical blood at birth. As derived from Figure 4, sampling after day 1 mainly occurred on days 3–4. Imaz et al. recommend sampling at birth, after 2 days (i.e., 48 h) and 1 week postpartum for all neonatal feeding trajectories, and additionally at 1 and 2 months in case of exclusive breastfeeding during maternal lithium use [44,45]. In addition, if neonatal lithemia is <0.2 mmol/L, an additional measurement is only recommended in case of symptoms [45]. Since our median neonatal lithium TDM only fell below 0.20 mmol/L from day 5 (Figure 4), we suggest follow-up at least until this moment, coinciding with the clinical observation period of 5 days. In summary, TDM sampling at birth, day 2–3, and day 5 is advised in our proposal. More frequent (i.e., daily) and longer (i.e., beyond day 5) monitoring of lithium TDM is relevant in case of observed or suspected (based on clinical symptoms) lithium toxicity. The need for other biochemical investigations (e.g., sepsis screening), technical investigations, as well as pharmacological NAS treatment, is based on an individual, case-by-case assessment. The local hospital protocol during the study recommended pharmacotherapy in case of a persistently elevated (modified) Finnegan score (3 times > 8), a strongly elevated score (>12), convulsions or severe dehydration. The topic of pharmacological NAS treatment will not be further discussed here but can be found in relevant reviews [51,52]. 

In the future, the proposed postnatal care protocol could be further modified according to local practices and preferences and updated considering new insights based on larger cohorts. The availability of a protocol may enhance uniformity and quality of care for future neonates with in utero exposure to lithium and can be used as a guide for parental counselling.

### 4.3. Strengths and Limitations 

Since in utero exposure to lithium is still relatively rare, clinical care recommendations for neonates are currently limited. Therefore, a broad range of parameters was examined providing a detailed overview of the early postnatal outcome and care of neonates in utero exposed to lithium. This evidence was supplemented with information from literature and used to develop a care proposal as a next step. 

However, the study approach has some limitations. First, and in spite of a study period covering 10 years in the largest Belgian University Hospital, the number of included cases still remained small. Second, no assessment was provided about long-term neonatal outcomes after in utero exposure to lithium, despite being at least as important as short-term outcomes. At present, only a few studies report on developmental outcomes. Forsberg et al. showed no significant association between mothers’ prenatal exposure to lithium or mood disorders and their offspring’s intelligence quotient [53]. Likewise, van der Lugt et al. reported no adverse effects on the growth, neurological, cognitive and behavioural development of children (3–15 years) with continuing lithium exposure during pregnancy [24]. However, both studies had a retrospective design and a small cohort [24,53]. The prospective studies of Schou (1976) and Jacobson et al. (1992) reported no differences in development between exposed children and controls [9,54,55], and very recently Poels et al. (2022) found no evidence for significantly altered neuropsychological functioning of children (6–14 years) with previously in utero exposure to lithium [56]. Third, the data were retrospectively collected, which means they were dependent on the quality of the registration of data in medical records.

## 5. Conclusions

This study retrospectively reports in detail the postnatal characteristics and short-term neonatal outcomes after in utero exposure to lithium. Since one case was diagnosed with NDI, we want to raise awareness of the occurrence of this rare adverse effect in this population. Due to the low incidence of in utero exposure to lithium, multicenter data pooling and structured long-term follow-up of these neonates are needed to further increase knowledge. To support clinical practice, a clinical postnatal care protocol for neonates with in utero exposure to lithium was provided. The median neonatal lithium TDM fell below 0.20 mmol/L (i.e., the level below which TDM is only recommended if symptomatic) from PNA day 5, which coincides with the suggested clinical observation period. Although this protocol may enhance the quality of care for future neonates with this condition and can be used as a guide in parental counselling, future research, including prospective protocol evaluation, is needed.

## Figures and Tables

**Figure 1 ijerph-19-10111-f001:**
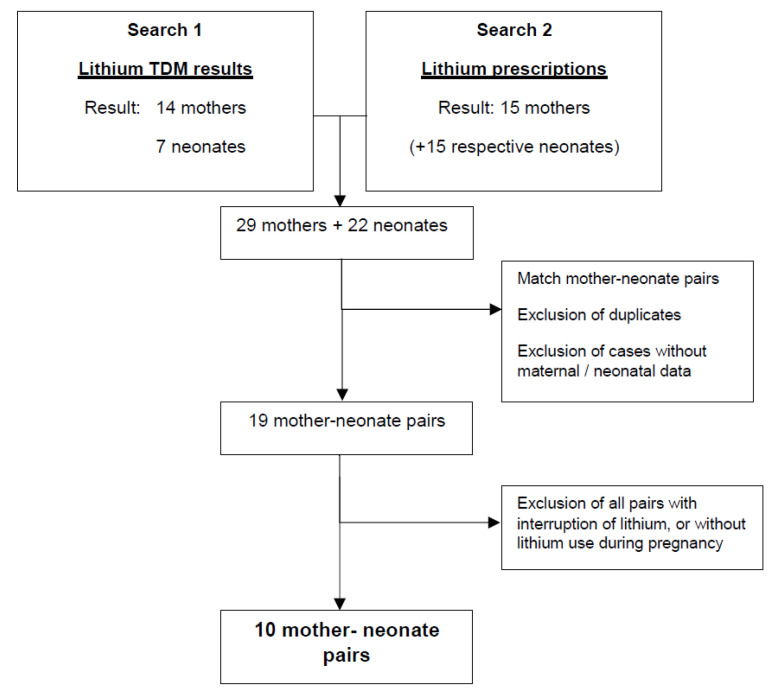
Flowchart presenting the recruitment of the mother-neonate pairs.

**Figure 2 ijerph-19-10111-f002:**
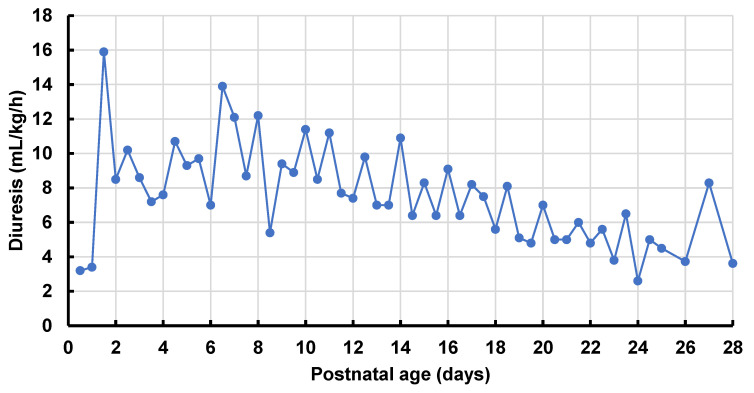
Detailed diuresis, expressed in mL/kg/h and assessed 12-hourly, during the neonatal period (i.e., first 28 days of life), of 1 neonate of the study cohort presenting with nephrogenic diabetes insipidus. Details on this case can be found in the text.

**Figure 3 ijerph-19-10111-f003:**
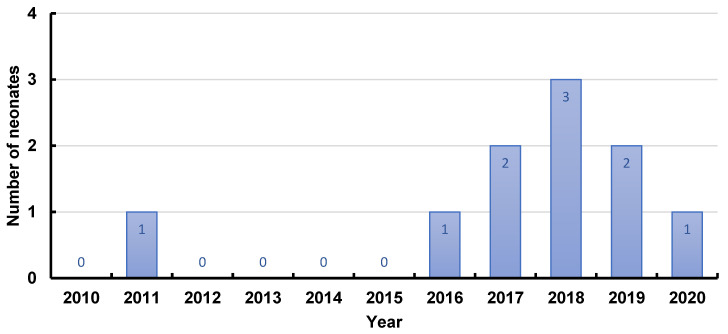
Incidence of neonates with in utero exposure to lithium, born at the University Hospitals Leuven, during the study period.

**Figure 4 ijerph-19-10111-f004:**
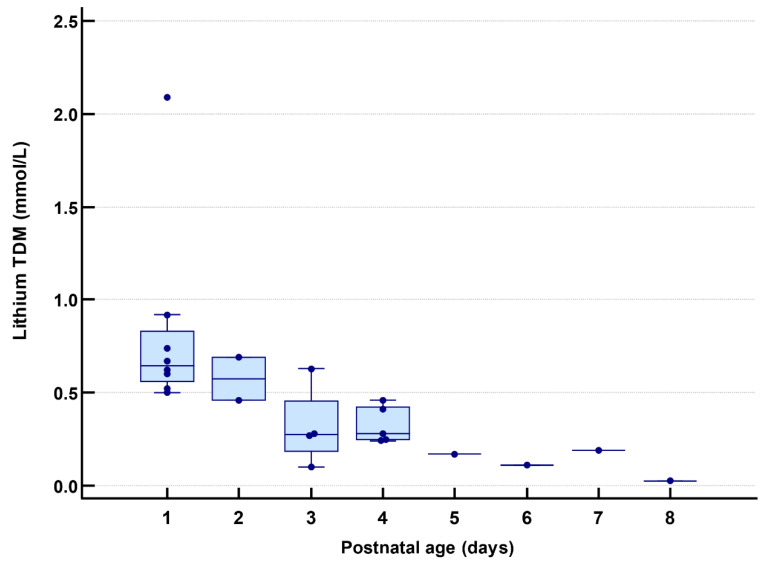
Available lithium therapeutic drug monitoring (TDM) data (mmol/L) of the included neonates are presented as daily boxplots covering postnatal age day 1 to day 8. One value reported as below the limit of quantification (LOQ), <0.05 mmol/L, was replaced by LOQ/2 = 0.025.

**Figure 5 ijerph-19-10111-f005:**
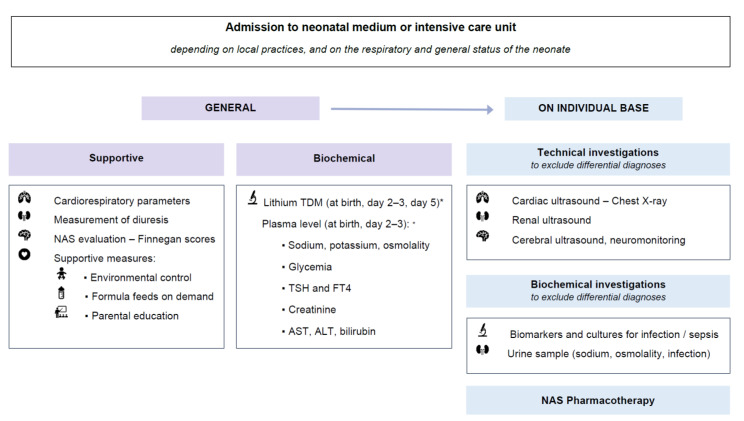
Proposal of postnatal care management of neonates with in utero exposure to lithium. NAS: Neonatal Abstinence Syndrome, TDM: therapeutic drug monitoring, AST: aspartate transaminase, ALT: alanine transaminase, FT4: free thyroxine, TSH thyroid stimulating hormone. * In case of observed/clinically suspected lithium toxicity more frequent (daily) and longer (beyond day 5) TDM is suggested. If TDM < 0.2 mmol/L, further sampling is only recommended in case of symptoms.

**Table 1 ijerph-19-10111-t001:** Characteristics of the included mothers.

Study ID Mother		Primary Diagnosis	Pregnancy Complications	Perinatal Lithium Dose (mg/day)	Perinatal Lithium Dose (mg/kg)	Lithium TDM at Delivery (mmol/L)	Concomitant Medication
1		Bipolar disorder	Hypertension	600	7.20	0.50	levothyroxine, quetiapine
2		Bipolar disorder	Gestational diabetes mellitus	1200	15.52	0.60	levothyroxine, quetiapine, acetylsalicylic acid, insulin, olanzapine
3		Bipolar disorder	Gestational diabetes mellitus	1500	20.32	0.84	levothyroxine, haloperidol, lamotrigine
4		Bipolar disorder	Gestational diabetes mellitus	1000	10.48	0.72	levothyroxine, lorazepam, olanzapine
5		Bipolar disorder	-	1400	11.57	0.64	aripiprazole, acetylsalicylic acid
6		Bipolar disorder	-	500	6.57	0.63	levothyroxine
7		Bipolar disorder	PPROM, shortened cervix	168	3.57	0.18	nifedipine
8		Bipolar disorder	-	625	9.08	0.54	levothyroxine
9		Bipolar disorder	-	1000	10.89	0.58	-
10		Bipolar disorder	Intrahepatic cholestasis of pregnancy Gestational diabetes mellitus	1000	10.47	0.92	levothyroxine, quetiapine, insulin
Median				1000	10.48	0.62	
IQR				600–1200	7.20–11.57	0.54–0.72	

TDM: therapeutic drug monitoring, IQR: interquartile range, PPROM: preterm pre-labor rupture of membranes, -: not applicable.

**Table 2 ijerph-19-10111-t002:** Characteristics of the included neonates at birth.

Study ID Neonate	Sex	GA (Weeks)	BW (Grams)	Apgar 1	Apgar 5	Apgar 10	pH *	Congenital Malformation
1	F	38	2990	5	6	9	7.25	-
2	F	37	3440	9	9	10	-	-
3	M	39	3010	9	9	10	-	-
4	F	37	2200	9	10	10	7.27	-
5	M	37	2620	9	10	10	7.23	-
6	F	36	2950	2	4	8	7.13	-
7	F	29	1200	6	4	5	7.31	CDH **
8	F	40	3100	9	9	9	7.21	-
9	M	41	4805	8	9	9	-	-
10	M	36	3505	10	10	10	7.15	-
Median		37	3000	9	9	9.5	7.23	
IQR		36–39	2620–3440	6–9	6–10	9–10	7.17–7.27	

F: female, M: male, GA: gestational age (only full weeks are reported), BW: birthweight; * arterial umbilical pH, ** CDH: congenital diaphragmatic hernia.

**Table 3 ijerph-19-10111-t003:** Characteristics of the included neonates during their hospital stay.

Study ID Neonate	Need for Respiratory Support	Duration of Respiratory Support (Days)	Diet	Time to Full Enteral Feeding (Days)	Time to Full Oral Feeding (Days)	Length of Stay (Days)	Lithium TDM at Birth (mmol/L)
1	Yes, CPAP	1	Formula	0	7	9	0.50
2	No	0	Formula	5	5	10	0.62
3	No	0	Formula	0	0	8	0.92
4	No	0	Formula	0	0	20	0.52
5	No	0	Formula	0	0	8	0.60
6	Yes, CPAP	1	Formula	3	3	8	0.67
7	Yes, IMV	37	Formula	-	-	37	-
8	No	0	Formula	0	0	6	2.09
9	No	0	Formula	0	0	6	-
10	No	0	Formula	0	0	12	0.74
Median		0		0	0	8.5	0.65
IQR		0–1		0–0.75	0–3.50	8–12	0.56–0.83

CPAP: Continuous Positive Airway Pressure, IMV = Invasive mechanical ventilation, -: no data available, TDM: therapeutic drug monitoring.

**Table 4 ijerph-19-10111-t004:** Biochemical (blood) data on postnatal age day 1, of the included neonates.

Study ID Neonate												
Biochemical Parameter	1	2	3	4	5	6	7	8	9	10	Median	IQR
Sodium (mEq/L)	135.50	141.80	-	139.00	138.20	138.30	139.20	137.90	139.50	136.70	138.30	137.60–139.28
Potassium (mEq/L)	6.03	5.16	-	6.18	5.73	4.49	4.52	5.67	5.28	4.91	5.28	4.81–5.81
AST * (U/L)	44.00	-	-	29.00	31.00	-	-	-	-	-	31.00	29.50–40.75
ALT ** (U/L)	6.00	-	-	5.00	6.00	-	<5.00	-	-	-	5.50	3.75–6.00
Creatinemia (mg/dL)	0.74	0.91	-	0.61	0.73	0.81	0.67	0.73	0.64	0.65	0.73	0.65–0.76
FT4 *** (pmol/L)	14.10	-	15.70	14.00	16.60	16.80	14.30	15.00	13.12 ^#^	15.60	15.00	14.08–15.93
TSH **** (mIU/L)	8.48	8.32	14.56	6.91	31.82	17.61	10.25	7.97	13.67	14.80	11.96	8.32–14.80
Glycemia, mean (mg/dL)	61.20	56.33	64.25	62.50	57.25	59.75	64.40	113.00	66.00	67.80	63.38	59.75–66.00
Glycemia, maximum (mg/dL)	73.00	62.00	82.00	70.00	73.00	60.00	112.00	113.00	66.00	80.00	73.00	66.00–82.00
Glycemia, minimum (mg/dL)	46.00	56.00	56.00	50.00	41.00	44.00	20.00	113.00	66.00	43.00	48.00	43.00–56.00

* AST: aspartate transaminase; ** ALT: alanine transaminase; *** FT4: free thyroxine; **** TSH: thyroid stimulating hormone, -: no data available. All values are provided with 2 decimal numbers. For the ALT value below the lower limit (<5.0), the lower limit/2 (=2.5) is used to calculate median and IQR. ^#^ For one neonate FT4 was reported in ng/dL (2011). Conversion to pmol/L was performed as ng/dL × 12.86.

**Table 5 ijerph-19-10111-t005:** Neurological assessment and lithium neonate/mother therapeutic drug monitoring (TDM) ratio.

Study ID Neonate	Highest Finnegan Score	Timing of Highest Finnegan Score	Neonatal Lithium TDM at Birth (mmol/L)	Neonate/Mother TDM Ratio	Clinical Neurological Examination, or Cerebral Ultrasound
1	6	Day 1	0.50	1.00	Floppy, irritable, mild tremor
2	4	Day 2–3	0.62	1.03	Sleepy, feeding difficulties
3	-	-	0.92	1.10	Normal
4	7	Day 3	0.52	0.72	Floppy, irritable, sleepy, feeding difficulties
5	-	-	0.60	0.94	Normal
6	4	Day 1	0.67	1.06	Floppy, horizontal nystagmus until day 2, slow grasping reflex, feeding difficulties
7	-	-	-	-	Grade 3 intraventricular haemorrhage
8	1	Day 3	2.09	3.87	Normal
9	-	-	-	-	Normal
10	3	Day 1–2	0.74	0.80	Normal
Median	4		0.65	1.02	
IQR	3–6		0.56–0.83	0.87–1.08	

IQR: interquartile range, TDM: therapeutic drug monitoring.

## Data Availability

The data presented in this study are available on request from the corresponding author.

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
