# Peer review of "Early Postnatal Outcome and Care after in Utero Exposure to Lithium: A Single Center Analysis of a Belgian Tertiary University Hospital"

_ijerph, 2022, doi:10.3390/ijerph191610111_

Round 1

Reviewer 1 Report

Congratulations, this is very detailed and holistic research. I would like address several considerations:

  1. The extensive research on neonatal condition after lithium exposure in utero is quite scarce, I hope you would do a prospective study about the long-term outcome of the affected neonates included in this study.
  2. The neonates diuresis scheme could be combined into one figure with special notes to the neonate with diabetes insipidus (perhaps, use different colour or dashed line)
  3. I understand the aim of this study is to describe the findings after prenatal lithium exposure, without comparing to the control group (non-exposed), perhaps it is more informative of we can see the comparison of Finnegan score, neonate lithium TDM, between each neonates in a form of plot rather than a table

Author Response

Dear Reviewers,

We sincerely thank you for the time taken to read our paper, and for the constructive and overall positive evaluation. We hereby would like to re-submit our paper entitled ‘Early postnatal outcome and care after in utero exposure to lithium: A single center analysis of a Belgian Tertiary University Hospital’ to the special issue on Pharmacotherapy during Pregnancy, Childbirth and Lactation in International Journal of Environmental Research and Public Health. The manuscript has been revised based on the suggestions of the reviewers. Below you can find a response to the comments.

Reviewer 2 Report

The manuscript presents a retrospective descriptive study of neonates born after in-utero exposure to Lithium, with a sample size of 10 monther-neonate pairs. The authors provide a detailed description of the cases followed by a proposed postnatal care protocol for neonates after in-utero exposure to Lithium. The manuscript is written in good quality english and cases are described in good detail. Parts of the manuscript appeat too wordy and could benefit from condensing it. 

Comments: 

1. Introduction: Can be shortened. The pathophysiology & mechanisms of action including maternal adverse effects could be condensed since this paper focuses on neonatal adverse events. Please mention current guidelines or recommendations for choice of medications for treating bipolar disorder during pregnancy. It may be pertinent to include it under introduction. 

2. Please consider condensing the figures and tables depicting results. Some results don't seem to be adding additional value - eg: diuresis of patients in figure 3. 

3. If you prefer to include the figures as is, please note error in figure 3 legend - "... presented separately in ...'. This should be figure 2, not figure 3. 

4. Figure 2 legend - please mention that this is one patient's data as described in the text. 

5. Table 5 listing finnegan scores is grossly inadequate due to missing data. May i suggest removing this table and describe in text. I would also recommend toning down the description of NAS scores since only limited data is available for the study patients. 

6. Figure 5 b: The significance of this data shown in the figure seems unclear. 

7. The authors recognize the limitations of the manuscript adequately. 

Author Response

(The authors gave the same response as above.)

Reviewer 3 Report

This manuscript deals with an important topic. But, several concerns need to be addressed as follows:

1.      The presentation of the material and methods section is a major concern in the current manuscript. The material and methods have been concisely presented without clearly describing the estimated parameters and statistical analysis. The authors should be divided this section with subheadings. How the lithium was measured? What is the statistical model used?

2.      There is a problem in using abbreviations throughout the manuscript. The full term should be mentioned first with the abbreviation between paresis then the abbreviations should be exclusively used throughout the manuscript. E.g., in line 100, free thyroxine has been abbreviated as T4, then the full term has been repeated in line 165. Such errors have been repeated for many abbreviations throughout the manuscript.

3.      The writing style should be formal from the third-person perspective. Do not use our (E.g. line 209) and we (E.g. line 209).

4.      The reference style in the text should be revised. E.g. line 59: Munk-Olsen et al (2018).

Author Response

(The authors gave the same response as above.)

Round 2

Reviewer 3 Report

No further comments to be addressed